# Language workbench user interfaces for data analysis

Victoria M. Benson and Fabien Campagne

The HRH Prince Alwaleed Bin Talal Bin Abdulaziz Alsaud Institute for Computational Biomedicine, The Weill Cornell Medical College, New York, NY, United States of America

Department of Physiology and Biophysics, The Weill Cornell Medical College, New York, NY, United States of America

## ABSTRACT

Biological data analysis is frequently performed with command line software. While this practice provides considerable flexibility for computationally savvy individuals, such as investigators trained in bioinformatics, this also creates a barrier to the widespread use of data analysis software by investigators trained as biologists and/or clinicians. Workflow systems such as Galaxy and Taverna have been developed to try and provide generic user interfaces that can wrap command line analysis software. These solutions are useful for problems that can be solved with workflows, and that do not require specialized user interfaces. However, some types of analyses can benefit from custom user interfaces. For instance, developing biomarker models from high-throughput data is a type of analysis that can be expressed more succinctly with specialized user interfaces. Here, we show how Language Workbench (LW) technology can be used to model the biomarker development and validation process. We developed a language that models the concepts of Dataset, Endpoint, Feature Selection Method and Classifier. These high-level language concepts map directly to abstractions that analysts who develop biomarker models are familiar with. We found that user interfaces developed in the Meta-Programming System (MPS) LW provide convenient means to configure a biomarker development project, to train models and view the validation statistics. We discuss several advantages of developing user interfaces for data analysis with a LW, including increased interface consistency, portability and extension by language composition. The language developed during this experiment is distributed as an MPS plugin (available at http://campagnelab.org/software/bdval-for-mps/).

Corresponding author
Fabien Campagne,
fac2003@campagnelab.org

## INTRODUCTION

Popular software available to analyze biological data is often developed with a variety of technology, including programming languages as well as standard and non-standard data formats used to configure the tools for specific analyses. While scientists with a broad computational background have no major difficulty using most of the software tools, this is

not the case for scientists whose training did not include computational experience, such as many biologists or physician scientists.

## Helping biologists take advantage of analysis tools

Several approaches have been used in the past to make it easier to use tools for biological data analysis. These approaches fall under the following broad categories:

1. **Custom graphical user interfaces.** User interfaces (UIs) can be programmed with a variety of technology to present the end-user with graphical user interfaces (GUIs) that simplify a specific analysis task. While earlier custom user interfaces were developed as desktop applications (e.g., Clustal-X (*Thompson et al., 1997*), Viseur (*Campagne et al., 1999*)), many biological analysis tools are now programmed and offered as web applications (e.g., from the TMHMM server (*Krogh et al., 2001*) to the vast number of tools described in the Nucleic Acids Research Web Server Special Issue).

2. **Generic workflow systems.** A minimal workflow system consists of a user interface to create and maintain workflows, and of a runtime engine to execute the workflows. A workflow consists of a set of elements representing units of work (often called workflow components), and connections among the elements representing how data flows from one element to another when the work of an element is completed. Examples of workflow systems include Taverna and Galaxy. These systems were developed to make it easier for biologists to assemble analyses pipelines that require using several tools. Supporting new tools in these systems does not require developing new user interfaces, but in exchange the ability to customize the system is limited to analyses that can be expressed with the workflow abstraction. The workflow abstraction has limited expressiveness. For instance, a workflow system does not support notions of variables or loops, which are often used in scripting or programming languages to automate repetitive tasks. While it is possible to use a scripting or general programming language to implement a work element and therefore perform repetitive tasks inside a workflow component, this requires knowledge beyond that of the workflow system, which the intended audience often does not have.

3. **Training workshops.** Most universities and medical schools offer training workshops to help biologists and clinical investigators learn the computational skills needed to work with command line analysis tools. Instructors who have taught such workshops know that most of the instructors' time is spent explaining technical aspects of the operating system and command line user interface rather than the concepts needed to perform a given analysis. Training workshops that focus on teaching workflow systems avoid this problem, but teach their trainees only how to solve the subset of problems that workflow systems can represent.

## Language workbench technology

Readers are refered to *Voelter et al. (2013)* and *Erdweg et al. (2013)* for an introduction to Language Workbench (LW) technology. We previously discussed the advantages of LW technology in Bioinformatics in the context of the development of GobyWeb

plugin scripts (*Simi & Campagne, 2014*). In this new study, we investigated whether LW technology—specifically the MPS Language Workbench (see http://www.jetbrains.com/mps/ and *Campagne, 2014*)—could be used to model the high-level concepts that an analyst should be aware of when he or she conducts an analysis. The MPS platform is one of several LW platforms that have been developed. Other platforms include Spoofax (*Kats & Visser, 2010*), Intentional Programming (*Simonyi, 1995*), or XText (*Eysholdt & Behrens, 2010*). We have chosen MPS for our study because of FC's increasing familiarity with the platform (*Campagne, 2014*).

The central innovation of the approach we present in this manuscript is to provide the end-user of an analysis tool with user interface elements that directly represent the high-level analysis concepts that the user needs to understand to perform a specific analysis. We will illustrate this idea in this manuscript with an analysis tool that helps end-users develop biomarker models from high-throughput data. We presented an introduction to the biomarker development and validation process in *Deng & Campagne (2010)*. Similar descriptions are also available in the MAQC-II articles (*Shi et al., 2010*). This manuscript follows the conventions and definitions given in this published work, which are briefly summarized here.

## The biomarker development process

Biomarker development studies measure a large number of molecular *features* across a collection of biological specimens, called *samples*. Data obtained for $k$ subjects and $n$ samples constitute a *dataset* in the form of a table of features. In the MAQC-II study, for instance, microarray assays were used to assemble 6 training datasets and 6 validation datasets. Several microarray *platforms* were used across the study. A total of 13 prediction *endpoints* were defined across these datasets that encoded a specific condition about each subject. For instance, in one dataset, an endpoint was defined by whether a subject of the breast cancer dataset was ER2 positive. Models were developed against this endpoint to try and predict whether the sample, for which gene expression data was available, was ER2 positive. Simply stated, an *endpoint* is what a predictive model is trained to predict. In general, an endpoint may have two or more categories, be a continous value, or a survival outcome for each subject of a dataset.

All these concepts are well represented and can be directly manipulated by the end-user in the analysis tool developed in this study.

## MATERIALS & METHODS

### MPS language workbench

We used version 3.1 of the Jetbrains MPS Language Workbench.

### BDVal

We adapted the BDVal Ant script to make it easier to use BDVal with configuration files automatically generated from the MPS project. This modified script is distributed with BDVal version 1.3+.

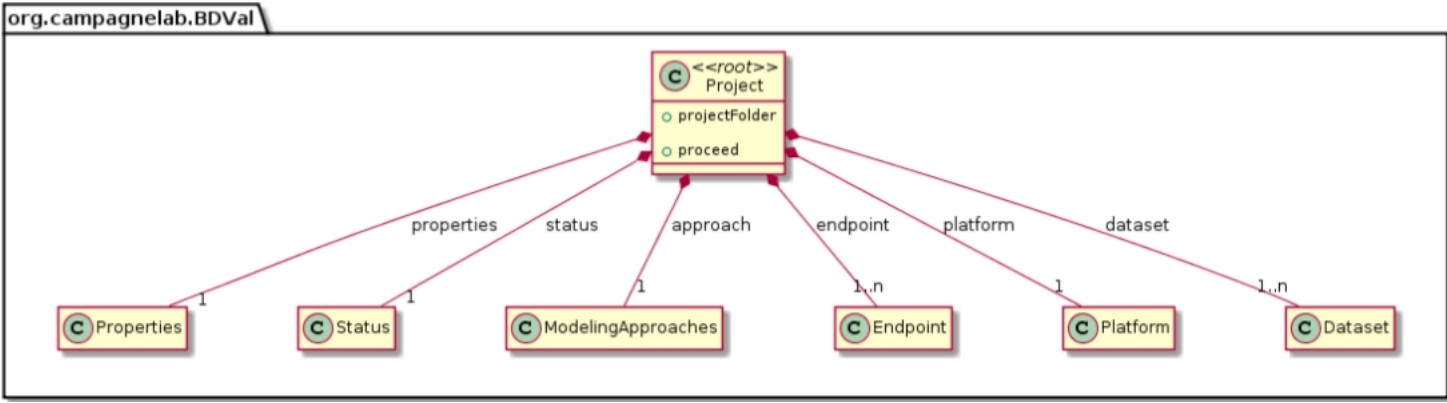

**Figure 1** **Main concepts of the BDVal MPS language.**

## Unified modeling language diagrams

The UML diagrams presented in this manuscript were generated automatically using the MPS language and the MPS UML_Diagrams plugin developed by our laboratory (https://github.com/CampagneLaboratory/UML_Diagrams).

## User interface buttons and file selection

Buttons and file selection buttons and dialogs were implemented using the MPS language org.campagnelab.ui (http://campagnelab.org/extending-mps-editors-with-buttons-just-got-easier/). The UI language was used to integrate buttons in the editor of the BDVal language concepts.

## Histogram charts

Charts were implemented with the org.campagnelab.XChart language, which integrates the XChart library with MPS (https://github.com/CampagneLaboratory/XChart).

# RESULTS

## Limitations of existing software

In a previous report, we described the BDVal software, which automates the development and validation of biomarker models from high-throughput data (*Dorff et al., 2010*). End-users of BDVal must configure a biomarker development project by creating a number of configuration files, as described in detail in the software online manual (*Chambwe & Campagne, 2010*). Configuring BDVal projects requires editing a number of files in different formats (including Ant build scripts written in XML, Java property files, and tab delimited files). Our experience suggests that this activity can be challenging and error-prone for non-technically savvy users.

## Modeling the biomarker development process

We modeled the biomarker development process with the MPS LW by creating concepts to represent each component of a biomarker analysis, as supported by the BDVal software (Fig. 1):

- **Project** This concept represents a biomarker discovery project and groups multiple datasets needed for analysis.
- **Sample** A sample is the biological material that was assayed. Each sample is associated with feature values measured by an assay.
- **Platform** The platform represents the assay used to measure features in the Sample. The platform concept links the assay probe Ids to the genes that the probe measures. Platforms are particularly important for micro-array based assays where multiple probes may measure the expression of the same gene.
- **DataSet** A dataset contains feature values for many samples.
- **Endpoint** An endpoint defines what the models will aim to predict. Endpoints are typically defined by the clinical application at hand (e.g., predict which patient respond to a treatment, or predict which patients have a specific disease).
- **Validation protocol** BDVal supports Cross Validation to estimate performance on a training set. Cross-validation can be performed with $k$ folds and with $r$ random repeats. The parameters $k$ and $r$ can be configured.
- **Feature selection approach** A feature selection approach implements a strategy to prioritize features according to their level of association with an endpoint. Several feature selection approaches have been implemented in BDVal and are available to reduce the subset of features used to construct a model. Feature selection is useful when there is a cost or time advantage in measuring few features values for each subject.
- **Classification approach** A classification approach is able to train a prediction model from feature values in each subject and information about the endpoint for these subjects. Several classification algorithms are implemented in BDVal, including Support Vector Machines with libSVM, Random Forests, Naive Bayes, LogitBoost and the K-Star classification approach.
- **Modeling approach** The combination of one or more feature selection approaches and a classification approach is a modeling approach and can train a predictive model from a training dataset.
- **Model** A model contains all the information needed to predict an endpoint when presented with feature values for a subject. BDVal models are written to a self-contained ZIP archive which contains the parameters of the model and threshold value for prediction of an endpoint (*Dorff et al., 2010*).

We were able to develop MPS concepts that almost directly represent each of these components of the biomarker discovery process. For clarity, we present these concepts as UML diagrams. Figure 1 presents the Project, Dataset, Endpoint and Platform concepts. Figure 2 presents the concepts used to capture the configuration of the user computer. Figure 3 presents how a modeling approach includes both a choice of Feature Selection, Classification method, and a list of models that can be generated with the approach. Figure 4 illustrates how we leveraged concept interfaces and inheritance to support auto-completion for different feature selection and classification approaches.

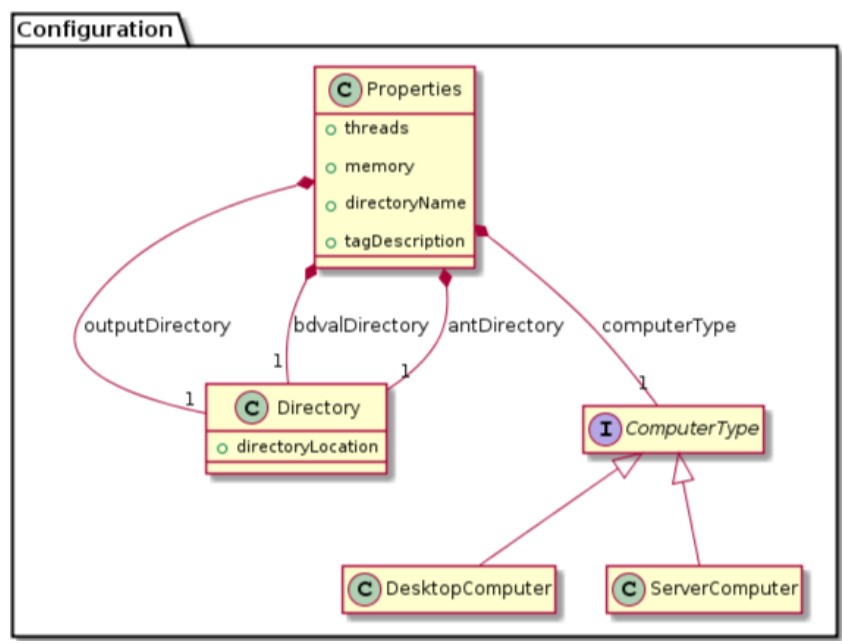

**Figure 2 Concepts for project configuration.** The Properties concept represents the configuration of the user computing environment. For instance, end-users can configure the project to use a server-class computer with at least 8 GB of memory and multiple cores, or a desktop computer with at least 1 GB of memory and a single core. This is achieved by selecting the appropriate ComputerType in the Project Properties. Project properties also define the location of two software packages that must be installed on the end-user's computer and the directory where models will be developed (outputDirectory).

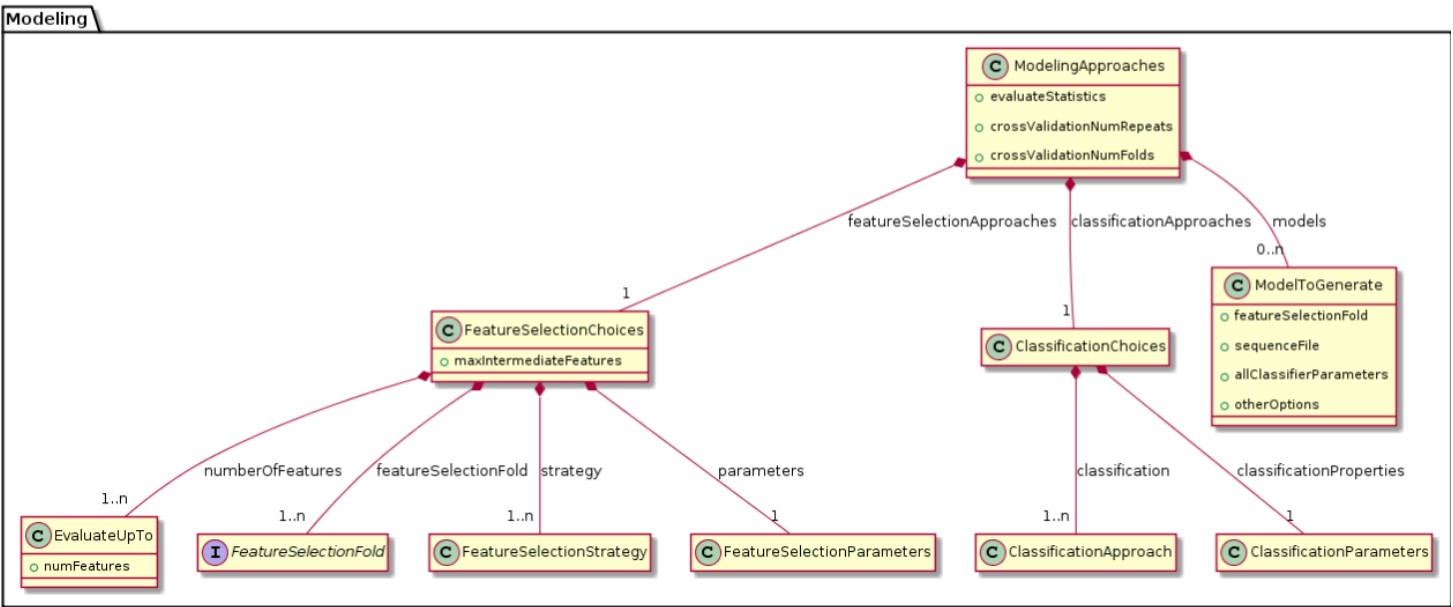

**Figure 3 Concepts for configuring modeling approaches.** BDVal makes it possible to combine choices for feature selection approaches and choices for classification approaches.

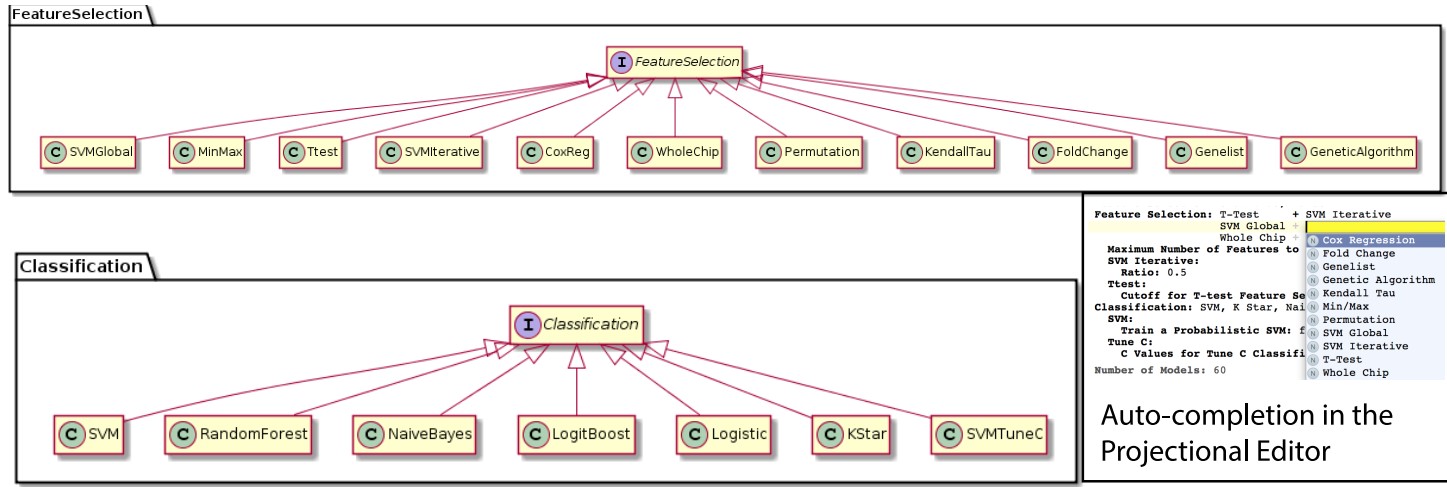

**Figure 4  Feature selection and classification approaches.** A FeatureSelection concept interface is implemented by individual concepts that represent each feature selection method supported by BDVal. The inheritance hierarchy is used by the MPS language workbench to offer auto-completion when an end-user needs to identify a feature selection approach (see bottom-right inset). The same mechanism is used to represent the classification approaches supported by BDVal.

Figure 4 presents concepts for the feature selection and classification approaches, which map directly to the methods supported by BDVal. Note the use of concept interfaces to indicate that each approach implementation can be used as element of the approach needed to construct a model.

## Generating and running a configured project

Instances of the Project concept can be configured by the end-user using the MPS user interface. When an instance is fully configured, it can be compiled within the LW. Compilation, also called generation in MPS, creates executable files from an instance of the Project concept and its children (instances of the concepts contained a Project). In the context of BDVal for MPS, generation converts the high-level analysis concepts described in the previous section to the configuration files needed to run BDVal on the end-user's machine. The generated BDVal files can be executed and will produce results files in configured output directory (a traditional user interface is generated for the Project to help provide feedback as the computation is progressing). The end-user can press the Refresh button (see Figs. 5 and 6, in the Status section) to trigger inspection of the output directory.

## Analysis user interface in a language workbench

Developing a user interface in a LW consists of creating an editor for each concept of a language. We developed editors for each concept to layout information in a logical manner. Figure 5 presents an instance of the Project concept rendered in the MPS projectional editor (*Campagne, 2014*). Project **Properties** are shown first because they need to be configured before a project can be executed (see Fig. 2).

**Project: Prostate_Example**

**Properties:**
  Desired Output Location:  /Users/fac2003/Downloads/test-bdval  …
  Location of BDVal Installation Directory:  /Users/fac2003/Downloads/bdval_1.3  …
  Location of ANT Installation Directory:  /Applications/apache-ant-1.9.4  …
  Type of Computer BDVal is running on:  Desktop
  Number of Parallel Threads to Use:  1
  Amount of Memory to Use (Mb):  1200
  Directory Name: FunFolder_1
  Tag Description: Testing Sample Prostate Project for BDVal Configured with MPS

**Approach:**
  Evaluate Statistics After Splits:  false
  External CV Repeats:  3
  External CV Folds:  5
  Number of Features:  20, 50, 100
  Feature Selection Fold:  true, false
  Feature Selection:  T-Test     + SVM Iterative                            │ optional: select mode
                     SVM Global  + optional: select second feature selection │ optional: select mode
                     Whole Chip  + optional: select second feature selection │ optional: select mode
    Maximum Number of Features to Keep After the First Step:  400
    SVM Iterative:
      Ratio: 0.5
    Ttest:
      Cutoff for T-test Feature Selection:  0.05
  Classification:  SVM, K Star, Naive Bayes, SVM Tune C
    SVM:
      Train a Probabilistic SVM:  false
    Tune C:
      C Values for Tune C Classification:  0.5, 1, 10
  Number of Models:  60

**Status:**   Refresh
  Result:
    Directory Name: FunFolder_1
    Folder Name: 20140903-1221-results
    Number of Models:  0

  20140902-1832-results

**Endpoints:** Fusion {YES, NO}

**Platform:** /Users/fac2003/Downloads/bdval_1.3/data/Prostate/GSE8402/platforms/GPL5474_family.soft.gz  …
  Type of Array:  Single Color Array
  Optional floor for the Signal Value:   optional: enter value

  Data Set:   enter name

    Run This Dataset:  true

    Prediction Endpoint :   select endpoint
      Categories :  select category

    Input:   enter file path …

**Figure 5  Project user interface overview.** This snapshot presents a view of a BDVal Project instance in the MPS editor. A project contains different sections: Properties, Approach, Status with Result folder, Platform (describing the assay and the datasets produced with this assay). The Dataset section is shown still incomplete to illustrate error highlighting in the MPS language workbench. The editor is fully interactive: end users can add datasets, feature selection methods, additional number of features. Some classification approaches have parameters that are only displayed when the feature selection method is part of the Project (e.g., the SVM Tune-C approach performs a parameter scan for the C cost parameter of a linear support vector machine). These features make the MPS editor a convenient user interface to configure a biomarker development project.

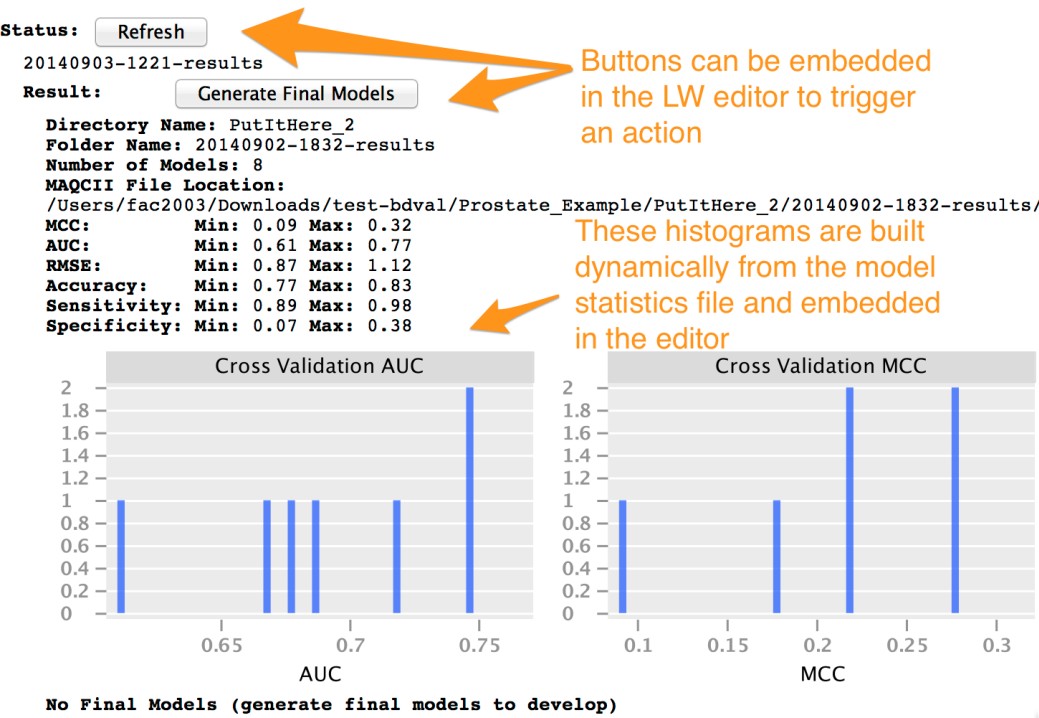

**Figure 6** The LW user interface can incorporate interactivity and graphics.

### Approach section

This section makes it possible to define which approaches should be used to develop and evaluate models. It includes the description of the cross-validation parameters (i.e., number of folds and number of random repeats) and provides an interactive editor to configure feature selection methods. The figure shows for instance an approach section configured to select features with a T-test followed by SVM iterative feature selection (recursive feature elimination), a simpler selection by SVM weights (SVM Global) and no selection where all the features (Whole Chip) are used to train a model.

The Approach section ends by showing the number of models. This number is calculated dynamically by considering the different feature selection, classification and parameters entered so far in the editor.

### Status section

This section offers a button to refresh the status. Pressing this button scans the output directory to identify results directories generated after BDVal is executed. Each directory found is shown with the folder name and the number of models that were generated by the execution.

### Endpoints section

This section makes it possible for the end-user to define one or more prediction endpoints, and the various possible categories for each endpoint. In the example shown,

the Fusion endpoint models the TMPRSS2-ERG gene fusion fusion event described in *Setlur et al. (2008)*.

### Platform section

This section indicates the location of the platform file in the GEO database soft format (*Edgar, Domrachev & Lash, 2002*). The Dataset section was intentionally left blank to illustrate that the content of Fig. 5 is not a plain text output, but provides error highlighting (red text is automatically highlighted by the MPS editor because the data attributes are required and missing).

We note that the LW user interface that we implemented is not limited to text. It includes user interface elements such as buttons that an end-user can press to perform some action, and can include images either static or dynamically generated at runtime. This is more evident in Fig. 6 where the histograms of the AUC and MCC statistics (estimated by cross-validation on the training set) are shown for a set of models developed with BDVal.

## Advantages of language workbench user interfaces

### UI consistency

We found that a user interface for data analysis implemented in a LW (LW UI) provides an important advantage over traditional approaches: the LW UI provides a consistent experience to the end-user. Custom developed UIs need to be designed for consistency and periodically reviewed to make sure that end-users have a predictable experience throughout the UI (*Nielsen, 1989*).

In this context, we note that the LW UI is defined using the *jetbrains.mps.editor* MPS platform language (see *Campagne, 2014*) and created from this definition by the LW platform. This mainly declarative approach to UI development ensures that most editing and navigation mechanisms provide a consistent user interface to the end-user.

For instance, auto-completion to set the value of a reference to other nodes already defined in the Project is an example of such a consistent UI behavior. Auto-completion is implemented by the LW platform and always provide a consistent user experience (which needs to be learned once and is then used used many times across LW UIs). Behaviors that are implemented consistently also include: copy and paste, source control, navigation to a referenced node, find usage, undo/redo.

We explain here the first two behaviors in more details:

1. A Project or a part of a project can be copied and pasted by the end-user. This is useful to make copies of a project before customization to explore a different analysis scenario. For instance, in BDVal, users may copy the Sample Project provided with the tool and start customizing this project to a new dataset and assay platform. With traditional technology for constructing user interfaces, copy and paste would need to be implemented explicitly by the programmer who creates the user interface. In a command line environment, copying a project would require copying a file, but copying a part of a project would require understanding the file format(s) for the file(s) that describe this part of the project. The LW UI provides a uniform mechanism for copy and

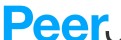

```
N  Prostate_Example  ×

   Project: Prostate_Example

   Properties:
     Desired Output Location: /Users/fac2003/Downloads/test-bdval ...
     Location of BDVal Installation Directory: /Users/fac2003/Documents/release-bdval_1.3 ...
     Location of ANT Installation Directory: /Applications/apache-ant-1.9.4 ...
     Type of Computer BDVal is running on: Desktop
     Number of Parallel Threads to Use: 1
     Amount of Memory to Use (Mb): 1200
     Directory Name: FunFolder_1
     Tag Description: Testing Sample Prostate Project for BDVal Configured with MPS

   Approach:
     Evaluate Statistics After Splits: false
     External CV Repeats: 3
     External CV Folds: 10   (1)
   ⬇ ⬆ ↺ 📋 📋  tures:  20, 50, 100
     External CV Folds: 5        true, false
(2)                  st      + SVM Iterative        optional: select mode
                 Fold Change + optional: select second feature selection  optional: select mode
                 SVM Global  + optional: select second feature selection  optional: select mode
                 Whole Chip  + optional: select second feature selection  optional: select mode
       Maximum Number of Features to Keep After the First Step: 400
       SVM Iterative:
         Ratio: 0.5
       Ttest:
         Cutoff for T-test Feature Selection: 0.05
(3)    Fold Change:
         Cutoff for Fold Change Feature Selection: 1.5
     Classification: SVM, K Star, Naive Bayes, SVM Tune C
       SVM:
         Train a Probabilistic SVM: false
       Tune C:
         C Values for Tune C Classification: 0.5, 1, 10
     Number of Models: 84
```

**Figure 7** Seamless integration with source control.

paste that does not require custom programming nor detailed understanding of the file formats affected by the operation.

2. The LW UI provides a tight integration with a source control system (Git, Subversion and CVS are supported). This makes it possible for the end-user to put a BDVal Project under source control to track changes to the analysis configuration. This is an important feature to support reproducible data analysis that requires no special effort from the LW UI developer. Figure 7 provides an illustration of how the MPS LW visually identifies changes made to a model after a commit, shows differences, and provides the ability to revert specific changes to the model. These features are offered by the platform consistently across all editors for the language, and do not require any custom coding. In contrast, such a source control integration of the software usage would be difficult to achieve with a traditionally developed user application. Developers of such an application would need to capture the state of the application in a file format, write the state to a file, call a source control tool to commit or checkout the application state, and provide a projection from the differences reported by the source control tool, into user interface elements. Because such developments are nontrivial, few if any custom data

**Peer**J

applications with user interface offer such a tight integration with source control. On the contrary, command line analysis tools integrate more naturally with source control: they process files that their users can put and maintain under source control. Source control over files reports differences at the level of granularity of a line, whereas LW UIs can pinpoint changes at different levels of granularity: from the level of a model property (e.g., see Fig. 7), all the way down to the insertion or deletion of a node of a model (e.g., the Project node for instance). Indeed, LW UIs track changes at the logical level established when designing the concepts of the analysis system. For instance, in BDVal for MPS, adding a feature selection approach to an analysis will record a single change to the Project, which will appear clearly as a change in feature selection when looking at revisions in the Difference tool. When the project is generated, this single change may affect several generated files, which if they were tracked in source control at their physical level would make it harder to recognize that these line-level changes are related.

### Language composition

A key feature of the MPS LW is to support seamless language composition (see *Voelter & Solomatov (2010)* and *Campagne (2014)*, Chapter 2). This feature is particularly useful in a data analysis context because the concepts that represent methods or datasets can be extended by the end-user very easily. For instance, in the context of BDVal, assume an end-user wanted to test the system with a number of random datasets, constructed with varying numbers of subjects and features with random values. With a custom user interface, it would not be possible to change the user interface of the tool to make it easier to configure such random datasets. Using the command line, it would be necessary to create simulated random datasets with the appropriate number of subjects and features, store these in separate files, and configure the project to use these new dataset files.

In a LW UI, we are able to create a new language that extends the BDVal language. This means that it is possible to define concepts and associated editors that provide different behavior, customized by the end-user. Assume we create such a language, and define the concept RandomDataset in this new language. We can declare that this concept extends Dataset (defined in the org.campagnelab.BDVal language). We can then define two attributes to encode the number of subjects and samples that the random dataset should have. In the RandomDataset editor, we provide the means to edit these parameters, and a button to create the dataset. The button would be pressed by the end user after he/she has configured the parameters. We can then define a behavior method in the RandomDataset concept that will construct the dataset file and configure the attributes of the super-concept. This behavior will be executed when the button is clicked. When this new language is imported into a model alongside the BDVal language, the end-user is able to create a RandomDataset instance in a BDVal Project, wherever it was previously possible to create a standard Dataset, and use it to configure random datasets. This extension mechanism does not require any modification to the original BDVal language. This simple scenario illustrates how language composition can be useful for data analysis.

Another way to extend the LW UI is to create new intentions for an existing language. Briefly, intentions are context dependent actions that the end user can activate after

positioning the cursor over some specific concept instance (see *Simi & Campagne (2014)* for illustrations, or *Campagne, 2014* Chapter 9). The MPS LW allows to define new intentions for concepts of another language. This feature can be used by a data analyst to automate common configurations. For instance, the BDVal language provides an intention that configures a Project with the location of the Sample data files distributed with BDVal. An analyst could define a language that provides intentions suitable to configure a BDVal project with specific analysis protocols (e.g., SVM only model construction, with predefined number of features).

### *Portability*

A LW UI is portable because it runs on the MPS LW, which is implemented in Java, and is available for Windows, Linux and Mac OS operating systems. A given analysis tool could limit portability to a specific platform. However, BDVal is also implemented in Java and the Java-based Ant build system, and will run on the same set of platforms as the MPS platform.

### *Distribution as language plugin*

We have created an MPS build script that generates an MPS plugin for the BDVal language. This plugin was uploaded to the LW plugin repository and has therefore become available to any user of the MPS platform. The MPS LW provides a convenient user interface to manage the discovery and installation of plugins. Different versions of a plugin can be offered and the user of a previous version is notified when a new version of an installed plugin has become available in the pugin repository. In a way, the MPS plugin repository can serve a similar function to the Galaxy ToolShed (*Blankenberg et al., 2014*).

While language plugins can depend on other plugins, it is currently not possible to specify a range of compatible versions for the target of a language dependency declaration (for instance, when specifying that the BDVal plugin depends on the XChart plugin, we are not able to indicate which range of versions of XChart is compatible with a given version of the BDVal plugin). This drawback is expected to limit the ability to evolve languages iteratively while maintaining compatibility with existing deployments.

## DISCUSSION

To our knowledge, our study is the first to propose and prototype the use of a Language Workbench to facilitate data analysis. Language Workbench technology has been developed in the context of general programming, and has been traditionally applied to developing programming languages that generate executable code. In this report, we have shown that the same technology can be successfully used to model the analysis process for biomarker development. Biomarker model development is an example of a complex biological data analysis that can benefit from high-level abstractions. Indeed, this activity requires using validation protocols such as cross-validation (*Stone, 1974*) or bootstrap (*Efron & Tibshirani, 1997*), and fully embedding feature selection in the cross-validation loop (or in the boostrap procedure) in order to avoid selection bias (*Varma & Simon, 2006*). Testing multiple feature selection approaches, machine learning methods, or parameters of these methods often produces tens of thousands of trained models across

splits of cross-validation and parameter choices. The large number of models and intermediate files (such as lists of features used to train the models) produced in these studies can quickly become a daunting data management challenge, even for analysts who know enough programming to automate analyses with a language such as R, Python or Java. The LW UI that we have described in this report facilitates the configuration of complete biomarker model projects, which can be executed to produce a large number of models in a single run. All models and intermediate files are automatically organized and, after completing, the LW UI displays performance evaluation statistics. The LW UI requires no programming experience and helps represent succinct solutions to biomarker development problems.

The idea of using a language workbench for data visualization (a component of data analysis) is not completely new. We identified precursors of this idea as early as 1975. For instance, a previous attempt was described as DDA (*Guthery, 1976*), a system designed to help with graphical data analysis. The DDA prototype was built with the LANG-PACK language design system (*Heindel & Roberto, 1975*). While a language design system, LANG-PACK was not a LW in the modern sense because it was based on parsing technology and therefore did not avoid the issue of ambiguity during language composition. While using similar wording to the present manuscript (i.e.,"language workbench," "interactive graphical data analysis techniques"), the DDA manuscript described a system to help draw lines on a screen for data visualization, which is a task orders of magnitude simpler than the biomarker data analysis task that we use as a motivating example in the present study.

We have also been conducting a large-scale test of the idea of using a language workbench for data analysis in related projects. The NYoSh analysis workbench, a data analysis platform constructed with the MPS language workbench, focuses on interactive analysis of high-throughput sequencing data and will be presented in detail in another article (http://workbench.campagnelab.org, Simi and Campagne, manuscript under preparation). The MetaR project (http://metaR.campagnelab.org) is a set of composable languages that help users create heatmaps from tables of data, or prepare other visualizations. Analyses written with MetaR generate to R code that can be executed directly within the MPS language workbench (Simi and Campagne, 2015, unpublished).

## CONCLUSIONS

In this manuscript, we propose that Language Workbench technology can be used to develop interactive user interfaces to help scientists with data analysis. We tested this idea by developing a prototype of a Data Analysis Language Workbench User Interface (DALWUI). This prototype was designed to help end-users develop and evaluate biomarker models from high-throughput datasets. This manuscript presented this prototype and briefly discussed the advantages of LW technology compared to custom graphical user interface development with traditional programming technology, or compared to command line driven data analysis.

Based on this experience, we propose that LW technology can be used to create a new generation of computational environment for data analysis and expect that these environments will yield key advantages for many data analysts, from the computational expert to the biomedical researcher with limited computational experience.

## ACKNOWLEDGEMENTS

The authors thank the developers of the Jetbrains MPS Language Workbench. We thank Manuele Simi for discussions and feedback on this manuscript. We thank the reviewers of this manuscript for constructive criticisms that helped clarify and strengthen this report.

### Funding

This investigation was supported by the National Institutes of Health NIAID award 5R01AI107762-02 to Fabien Campagne and by grant UL1 RR024996 (National Institutes of Health (NIH)/National Center for Research Resources) of the Clinical and Translation Science Center at Weill Cornell Medical College. The funders had no role in study design, data collection and analysis, decision to publish, or preparation of the manuscript.

### Grant Disclosures

The following grant information was disclosed by the authors:
National Institutes of Health NIAID: 5R01AI107762-02.
National Institutes of Health (NIH)/National Center for Research Resources: UL1 RR024996.

### Competing Interests

Fabien Campagne serves as an Academic Editor for PeerJ. He is the author of the book "The MPS Language Workbench, Volume I" and receives royalties from the sale of this book.

### Author Contributions

- Victoria M. Benson performed the experiments, reviewed drafts of the paper.
- Fabien Campagne conceived and designed the experiments, contributed reagents/materials/analysis tools, wrote the paper, prepared figures and/or tables, reviewed drafts of the paper.

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
