# Peer review of "Language workbench user interfaces for data analysis"

_PeerJ, doi:10.7717/peerj.800_

## Round 0.1 · original submission · Major Revisions

· Academic Editor

Major Revisions

The main concerns of the reviewers relate to the lack of context in this paper. Both reviewers expressed concern regarding claims made relative to existing tools such as Taverna and Galaxy. I agree that these tools were not sufficiently well discussed. I also found that this paper was a bit hard to read, due to an excessive reliance on the authors' previous work. In addition to revisions aimed at the reviewers concerns, this paper would also benefit from additional introductory material and perhaps a worked example showing how an appropriate workflow might be realized.

·

Basic reporting

The article is well written and provides appropriate references to prior literature.
Note: at line 74, the word "ant" should be replaced with "Ant" since it is a noun of a tool.

Experimental design

No comments

Validity of the findings

No comments

Additional comments

The article introduces an original and very interesting approach to build interactive data-analysis tools that provides the expressiveness of programming languages with the ease of graphical user interfaces.

However I find inappropriate the use of term "Dataflow" in the paper.

The authors classify Taverna and Galaxy as "Dataflow systems" (line 28), but both these tools define themselves as "workflow tool/system". See 1 and 2.

It seems they used the term "dataflow" in place of "workflow" or as synonyms. In my opinion this can be misleading since these terms identify two different computing paradigms.

In computer science "Dataflow" refers to a declarative computing model for concurrent tasks executions in which functions are evaluated over partial data structures by using Dataflow variables. Dataflow has been implemented both in hardware architectures (as alternative to traditional Von Neumann architecture) and programming languages. See: 3, 4 and 5.

I would suggest to change the term "dataflow" with "workflow" or split it in two separate words to avoid any possible ambiguity.

Also the authors state that "biomarkers model cannot be directly expressed with the dataflow model", but they do not explain why in the discussion (the mere fact that does not exist a plugin for Galaxy does not mean that it cannot be implemented with it or another workflow system).


1) The Taverna workflow suite. http://nar.oxfordjournals.org/content/41/W1/W557
2) Galaxy: a comprehensive approach for supporting accessible, reproducible, and transparent computational research. http://genomebiology.com/2010/11/8/R86
3) Advances in dataflow programming languages. http://dl.acm.org/citation.cfm?id=1013209
4) Programming languages for distributed computing system. http://citeseerx.ist.psu.edu/viewdoc/summary?doi=10.1.1.145.7873
5) Concepts, Techniques, and Models of Computer Programming. http://www.info.ucl.ac.be/~pvr/book.html

Reviewer 2 ·

Basic reporting

The manuscript is generally clear and very well written. However, I have a few critical remarks that must be addressed before the manuscript can be considered acceptable for publication.

Experimental design

The manuscript describes a software or language. The breast cancer study has been published elsewhere and is only used for illustration in this work.

Validity of the findings

The authors claim already in the abstract that "developing biomarker models from high-throughput data is a type of analysis that cannot be directly expressed with the dataflow model". Later, in the discussion, they claim that "Galaxy does not offer plugins suitable to develop biomarker models". This is simply false. A quick search reveal there are quite a few workflows, plugins and frameworks for Galaxy (and also for Taverna) precisely for developing biomarker models, using for example gene expression or mass spectrometry data. Dataflows for "biomarkers" with separate discovery/cross validation phases are not that hard to implement in e.g. Taverna or Galaxy (it has been done). The statistical language R is frequently used in the analysis of gene expression, proteomics and metabolomics data, also for "biomarker" discovery and validation. All specialized R packages, including SVMs and the other classifiers mentioned by the authors, are easily and directly accessible from within a Taverna workflow. The authors need to be much more specific on what limitations they refer to in the "dataflow model" of existing workflow management systems or languages. The authors must prove (or provide a reference) that a particular problem cannot be expressed in, say, t2flow before making such statements. I think this would be quite difficult, given that the Galaxy and Taverna folks have been using their workflow managers for biomarker discovery and validation projects for some time and quite a lot is published on the subject.

On p. 8, the authors claim "source control is difficult to integrate in a custom user interface and consequently is rarely used" and that "analysis with command line tools often takes advantage of source control systems, and represent changes at the level of granularity of a line". It is not at all clear to me that the development and maintenance of a user interface that embeds or calls many separate functions or external programs cannot be managed quite just as effectively with, say, GitHub. Alternatively, workflows can also version control, check out and build software used downstream in the same workflow. I suspect the LW UI is just another, possibly elegant, way of addressing the same problem. It is also not entirely clear what the authors refer to with the command line tools making advantage of version control systems.

In conclusion, the authors should carefully consider all statements comparing the LW model with other models, including proof that a particular task cannot be expressed with any dataflow model, if this is indeed the case.

---

## Round 0.2 · accepted · Accept

· Academic Editor

Accept

The reviewers have agreed that this revision addresses the issues identified in the first round of reviews. I agree with their assessment.

·

Basic reporting

No comment

Experimental design

No comment

Validity of the findings

No comment

Additional comments

The authors agreed on the prosed changes.

Reviewer 2 ·

Basic reporting

OK

Experimental design

OK

Validity of the findings

OK

Additional comments

The authors have successfully addressed the questions/concerns from both reviewers. I would also be willing to agree with the authors in their rebuttal, but think it is wise to be careful and specific when making claims about limitations in other models or software, especially when you are in control of delineations and able to impose certain limitations yourself. I am happy with the changes to the manuscript.